# Authentication of the Bilberry Extracts by an HPLC Fingerprint Method Combining Reference Standard Extracts

**DOI:** 10.3390/molecules25112514

**Published:** 2020-05-28

**Authors:** Bingbing Liu, Tiantian Hu, Weidong Yan

**Affiliations:** 1Department of Chemistry, Zhejiang University, Hangzhou 310027, China; 11637060@zju.edu.cn (B.L.); 3130101576@zju.edu.cn (T.H.); 2State Key Laboratory of Environmental and Biological Analysis, Hong Kong Baptist University, Kowloon Tong, Kowloon, Hong Kong 999077, China; 3Zhejiang Skyherb Biotechnologies Co., Ltd., Anji 313300, China

**Keywords:** bilberry extract, anthocyanins, HPLC, fingerprint, authentication

## Abstract

A simple and fast high-performance liquid chromatography (HPLC) fingerprint method combining reference standard extract for the identification of bilberry extract was developed and validated. Six batches of bilberry extract collected from different manufactures were used to establish the HPLC fingerprint. Other berry extracts—such as blueberry extracts, mulberry extracts, cranberry extracts, and black rice extracts—were also analyzed for their HPLC chromatograms. The fingerprints of five batches of bilberry extract showed high similarities, while one batch was distinguished from others. Additionally, the content of anthocyanin Cyanidin-3-*O*-glucoside (Cy-3-glc) in each berry extract was analyzed and compared. The results indicate that this HPLC fingerprint method, combining reference standard extracts, could be used for the authentication and quality control of bilberry extracts.

## 1. Introduction

*Vaccinium myrtillus* L., also known as the bilberry, is a small shrub belonging to the *Ericaceae* family. It is a very diverse species which distributes in the northern hemisphere [1]. Bilberries are rich in anthocyanins, which are proved to be free radical scavengers and antioxidants [2,3,4,5], thus making them a good source for natural health-care foods. Anthocyanins are found with anti-inflammation, anti-tumor, and vision-improving activities [6,7,8]. Besides, it can aid the regulation of blood lipid levels and improve insulin resistance [3,6,7]. More importantly, bilberries contain a higher proportion of anthocyanins than other berries, such as the blueberry, mulberry, strawberry and cranberry [9,10,11,12]. Under these circumstances, the bilberry extract industry is starting to show strong momentum [13]. Bilberry extract has been recognized as a non-certified food additive by the US Food and Drug Administration (FDA), with US sales reaching USD 3.75 million in 2012 [14].

The price of bilberry extracts in the market has been soaring, but quality is uneven. The adulteration of cheaper anthocyanin-rich extracts such as blueberry extracts, mulberry extracts, and black rice extracts has been found. Thus, a feasible and reliable method for the authentication of bilberry extracts is highly required.

Currently, research on anthocyanin in bilberries is still in the preliminary stage. The quantification of anthocyanin in bilberries and other plant extracts usually uses the ultraviolet spectrophotometry (UV) method and the HPLC (high-performance liquid chromatography) method [15]. The UV method has poor reproducibility and is susceptible to interference. Besides, the anthocyanin species cannot be identified by the UV method. Thus, new technologies for the identification and quantification of anthocyanins in plants and plant extracts have come into sight. For example, the analytical characterization of anthocyanins in bilberry fruit and food products was achieved by HPLC, coupled with mass spectrometry (MS), in 2018 [16]. Besides, it was found that the content of the anthocyanins of bilberry was influenced by several conditions, such as plant species, place of origin, climate, and freshness of the fruit. For example, the anthocyanin content of wild bilberries is generally higher than that of cultivated bilberries.

The HPLC fingerprint, combining standards, is a comprehensive method for the assessment of the quality and reliability of food and plant extracts [17,18,19,20]. The US FDA has suggested that herbal products should be evaluated by this strategy [21]. However, this requires a dozen of standards, since some standards are expensive and sometimes not available in the market, due to the difficulty of their preparation. On the other hand, the HPLC-mass spectrometry (MS) approach can identify components, but its accuracy and precision are inferior to the HPLC method; furthermore, the HPLC-MS instrument is expensive and more complicated than that of HPLC, requiring a professional background of its users [22].

Recently, our lab reported an HPLC method combining reference extracts, instead of individual standards [23], which makes up the deficiency mentioned above. The reference standard extract is used as a reference substance to identify and quantify the components of plant extract samples. Compared to the external standard method, it only requires standardization of the multiple markers of the reference standard extract in advance, which makes it economical and easy for the identification and quantification of plant components. The convenience and consistency of this method was validated in-house and its results showed no difference with that of the conventional external standards method [23]. Thus, this study aimed to develop an HPLC fingerprint method, combining reference standard extracts, for the authentication of bilberry extracts from other berry extracts, such as blueberry extracts, mulberry extracts, cranberry extracts, and black rice extracts. In addition, with only one standard obtained, we further quantified and compared the characteristic component Cyanidin-3-*O*-glucoside (Cy-3-glc) in each bilberry extract.

## 2. Results and Discussions

### 2.1. Optimization of the Elution Program

The separation resolution of each peak, as well as the total HPLC analysis time, was used to evaluate the fitness of the elution program. The separation resolution of each peak was obtained using Wufeng HPLC analysis software. Though elution program 2 (P2) had the shortest analysis time, the separation resolution of each peak was not proper, with some peaks overlapped and difficult to identify. It was found that the separation resolution of all the anthocyanins was best under elution program 6 (P6) (Table 1, chromatograms are shown in Appendix A), so this was set as the optimum elution program for further use.

### 2.2. Similarity Analysis of HPLC Fingerprints of Bilberry Extract Samples

As seen in Appendix A, five batches of bilberry extracts from different cultivating regions share the same anthocyanin profiles according to the HPLC chromatogram, and one batch has the unique chromatogram.

The similarities between the entire chromatographic profiles of the six batches of bilberry extract samples and the reference standard were calculated by the professional software Similarity Evaluation System for Chromatographic Fingerprint of Traditional Chinese Medicine (Version 2004A) [24]. The correlation coefficients of fingerprints from six batches were shown as 0.998, 0.993, 0.994, 0.997, 0.983 and 0.163. The results indicate that the five batches of bilberry extract samples from different plantations shared nearly the same correlation coefficients of similarities and one was different from others.

### 2.3. Establishment of the Bilberry Extract HPLC Fingerprint

HPLC fingerprints of six batches of bilberry extracts from different manufactures were obtained (representative chromatogram is shown in Figure 1B and detailed chromatograms are listed in Appendix A). Since elution program 1 in this study was the same as the elution program of international commercial standard for bilberry extract SW/T 2-2013 [25], 10 characteristic peaks of the bilberry reference extract were identified according to international commercial standard directly, without preliminary identification of the reference standard extract. The HPLC fingerprints of six batches of bilberry extract from different regions were obtained, selected, and matched by the Similarity Evaluation System for Chromatographic Fingerprint of Traditional Chinese Medicine (Version 2004A). Among these, five batches were in good accordance with that of the reference extract (Figure 1A), but one batch (A6) only showed one characteristic peak (Figure 1C), indicating that it might not be the genuine bilberry extract and needs further inspection by other technologies, such as HPLC-MS.

The retention time and peak area of the bilberry reference extracts are shown in Table 2, in addition to the relative retention time (RRT) and relative peak area (RPA) of the five similar bilberry extract samples, with respect to the reference peak (peak 5), which were compared by calculating the relative standard deviation (RSD). Good RSDs of the RRT and RPA (Table 2) revealed that these samples had similar HPLC profiles, indicating that the characteristic HPLC fingerprint of bilberry extracts was applicable.

This HPLC fingerprint method, combining reference standard extracts, is fast and easy and unusual samples can be subject to further inspection. In terms of identification, compared with methods such as HPLC combining external standards, it does not require the purchase of external standards, nor the laborious standards preparation procedure. For example, Muller used 15 anthocyanin standards to study the anthocyanin content of bilberry juices by HPLC-UV [26], the performance of which requires time to collect and prepare the standards. In terms of quantification, the HPLC fingerprint method combining reference standard extracts can provide accurate quantification of the components in samples. Since, in general, components of the reference standard extract are standardized before using as the reference standard for quantification [27,28,29,30]. The use of reference standard extract relieves the pressure of increasing demands for pure marker compounds [23]. 

### 2.4. HPLC Chromatograms of the Other Extract Samples

Then, HPLC chromatograms of the other extract samples are obtained. Seven batches of blueberry extracts were divided into two groups, where two batches showed chromatogram type as in Figure 2A and the other five batches showed chromatogram type as in Figure 2B; however, these chromatograms are quite different from that of the bilberry extract fingerprint. The different chromatograms of blueberry extracts obtained may be because of several factors, such as genotypes, cultivation regions, soil quality, ages, tree age, and harvest periods, all of which could affect the anthocyanin content of blueberries.

Likewise, four batches of mulberry extracts were analyzed; they all showed the same chromatograms (Figure 3), which were different from that of the bilberry extract fingerprint.

Chromatograms of the two batches of cranberry extracts were divided into two groups (Figure 4A,B), which were also different from bilberry extracts.

Finally, four batches of black rice extract were analyzed; they all showed the same chromatograms (Figure 5), which were also different from bilberry extracts.

By comparing the HPLC chromatograms of the bilberry reference standard extract with those of the blueberry, mulberry, cranberry, and black rice extracts, it can be concluded that the anthocyanin profile of bilberry extract is quite different from that of the other berry extracts. The bilberry extract is rich in anthocyanins, containing more than 10 kinds of anthocyanins, while other berry extracts only contain one to four kinds of anthocyanins. Therefore, this HPLC fingerprint method can effectively distinguish the bilberry reference standard extract from other plant extracts.

### 2.5. Validation of the HPLC Quantification Method

The linearity regression curve for Cy-3-glc was *y* = 29430.14143*x* − 0.37173, *r*^2^ = 0.9988. The linearity range was 0.02 mg/mL–0.12 mg/mL. The LOD (limit of detection) was 0.007 mg/mL, and the LOQ (limit of quantification) was 0.02 mg/mL.

Good precision, as revealed in the relative standard deviations (RSDs) for peak areas, was found (Table 3).

For the stability test, the RSDs of the peak areas of analytes were < 0.34%, indicating that the standard solution was stable for 24 h at ambient temperature.

The repeatability of the method was tested by the determination of a standard solution of Cy-3-glc. The RSDs of peak areas were calculated, which fell within the ranges specified, as shown in Table 4.

The recovery test was conducted to evaluate the accuracy of this method. The recovery test of the Cy-3-glc standard solution was obtained by adding a known amount (479.04 μg) of Cy-3-glc standard solution to the six bilberry extract sample solutions. As shown in Table 4, the recovery rate for Cy-3-glc was within the range of 99.61% and 102.76% and the RSD for recovery rate was 1.29%.

### 2.6. Quantification of the Cy-3-glc in Bilberry Extract Samples

The proposed HPLC method was successfully applied to the quantification of anthocyanin Cy-3-glc in bilberry extract samples, using the external standard calibration method. The contents of Cy-3-glc in bilberry extract samples were calculated as follows,
$$\eta =\frac{C_{x}}{C_{m}}$$
where *C_x_* is the concentration (mg/mL) of Cy-3-glc in bilberry extract sample solution, *C_m_* is the concentration (μg/mL) of bilberry extract solution, and *η* is the content of Cy-3-glc in bilberry extract sample.

As shown in Table 5, the contents of Cy-3-glc in bilberry extract samples were within the range of 62–77 μg/mg, with a slight difference.

### 2.7. Comparison of the Cy-3-glc Content in the Other Extracts

The Cy-3-glc contents in several batches of bilberry extracts, blueberry extracts, mulberry extracts, cranberry extracts, and black rice extracts were determined and presented in a whisker plot. A whisker plot is suitable to show the quantitative differences between the extracts. It can be clearly seen in Figure 6 that extracts from different origins had different Cy-3-glc contents for the blueberry extract, mulberry extract, cranberry extract and black rice extract samples, which were in the range of 8.3–594.5 μg/mg. Meanwhile, in bilberry extract samples, the Cy-3-glc contents were similar, which were in the range of 62–77 μg/mg. According to the HPLC analysis, these extracts contained fewer kinds of anthocyanins, where Cy-3-glc was the main anthocyanin component. As for cranberry extracts, these contained minimal Cy-3-glc content: 18.7–33 μg/mg.

## 3. Materials and Methods

### 3.1. Chemicals and Reagents

HPLC-grade acetonitrile and methanol were purchased from Sigma-Aldrich (St. Louis, USA). All other chemicals and reagents were of analytical grade and obtained from Sinopharm Chemical Reagent Co. Ltd. (Shanghai, China). Bilberry reference standard extract (European Pharmacopoeia reference standard) and cyanidin-3-*O*-glucoside chloride (98%) was purchased from J&K Scientific Ltd. (Beijing, China). The information on berry extracts is listed in Table 6.

### 3.2. Sample Preparation

#### 3.2.1. Preparation of Cy-3-glc Standard Solutions

Stock standard solutions of Cy-3-glc were prepared by dissolving 10 mg of cyanidin-3-*O*-glucoside chloride into a 25 mL volumetric flask with HCl/methanol (2:98, *v*/*v*) to volume, mixed. Of the stock solution, 0.25, 0.5, 1.0, 1.5, 2.0, 2.5, and 3.0 mL was transferred to a 10 mL volumetric flask and diluted with 10% aqueous phosphoric solution. Thus, a series of standard solutions, at concentrations of 0.01, 0.02, 0.04, 0.06, 0.08, 0.10, and 0.12 mg/mL, were obtained. Each standard solution was prepared every day and passed through a 0.45 μm PTFE filter before HPLC injection.

#### 3.2.2. Preparation of Reference Extract Solutions

Stock solutions were prepared by dissolving 125 mg of bilberry reference extract into a 25 mL volumetric flask with HCl/methanol (2:98, *v*/*v*) to volume, mixed. Of the above solution, 2 mL was accurately transferred into a 10 mL volumetric flask, diluted with 10% aqueous phosphoric solution prior to the HPLC analysis. Each sample solution was passed through a 0.45 μm PTFE filter prior to the HPLC injection.

#### 3.2.3. Preparation of Sample Solutions

The bilberry extract sample solution was prepared by preparing 125 mg of bilberry extract in HCl/methanol (2:98, *v*/*v*) in 25 mL flasks; 3 mL of this solution was transferred into a 10 mL volumetric flask, diluted with 10% aqueous phosphoric solution, prior to the HPLC analysis. Each sample solution was passed through a 0.45 μm PTFE filter, prior to the HPLC injection. Other berry extract samples were prepared by the same procedure indicated above.

### 3.3. Screen of the Chromatographic Elution Program

The HPLC analysis was performed on a Wufeng HPLC unit (Wufeng Co., Shanghai, China), coupled with an ultra-violet detector (1290 VWD). Separation was achieved on a Hedera ODS-2 C18 column (4.6 mm × 250 mm, 5 µm; Nacilai Tesque Inc., Japan). The detection wavelength was set at 535 nm and the flow rate was 1 mL/min. The injection volume was 20 μL. The injections were performed in triplicate for each sample.

According to the European Pharmacopeia, eluent A was 8.5% aqueous formic acid solution, and eluent B was acetonitrile/methanol/formic acid/water = 22.5:22.5:8.5:41.5 (*v*/*v*/*v*/*v*). Six gradient elution programs were performed, and the one with the optimum resolution was set as the optimum elution program. 

Program 1 (P1): 0–35 min: 7% B–25% B, 35–45 min: 25% B–65% B, 45–46 min: 65% B–100% B, and 46–50 min: 100% B for 10 min.

Program 2 (P2): 0–15 min: 7% B–25% B, 15–25 min: 25% B–65% B, 25–26 min: 65% B–100% B, and 26–30 min: 100% B for 5 min.

Program 3 (P3): 0–5 min: 7% B–25% B, 5–25 min: 25% B–55% B, 26–35 min: 55% B–100% B, and 36–40 min: 100% B for 5 min.

Program 4 (P4): 0–5 min: 7% B–25% B, 5–35 min: 25% B–55% B, 36–45 min: 55% B–100% B, and 46–50 min: 100% B for 5 min.

Program 5 (P5): 0–5 min: 7% B–25% B, 5–35 min: 25% B–45% B, 36–45 min: 45% B–100% B, and 46–50 min: 100% B for 5 min.

Program 6 (P6): 0–5 min: 7% B–25% B, 5–35 min: 25% B–40% B, 36–45 min: 40% B–100% B, and 46–50 min: 100% B for 5 min.

The sample solutions were prepared with the authentic bilberry reference extract.

### 3.4. Establishment of the HPLC Fingerprint of Bilberry Extracts

To establish the representative HPLC chromatogram, six batches of bilberry extracts, as well as the reference bilberry extract, were analyzed under the optimized HPLC condition. Then, seven batches of blueberry extracts, four batches of mulberry extracts, two batches of cranberry extracts, and four batches of black rice extract samples were analyzed, and their HPLC chromatograms were compared.

### 3.5. Validation of the HPLC Quantification Method 

According to the guidelines for the validation of drug quality standards of the China Food and Drug Administration (CFDA) [31], the linearity regression curves for each component were obtained by plotting the peak areas (y) against the concentrations of the Cy-3-glc standard solution. The limit of detection (LOD) was determined by considering a signal to noise ratio (S/N) of 3 and limit of quantification (LOQ) was determined by considering a signal to noise ratio (S/N) of 10. The precision of the HPLC method was evaluated by analyzing the replicated Cy-3-glc standard solution of the same sample (0.08 mg/mL) for three times within one day and on two consecutive days. The stability test was determined by sampling the same standard solution in 2, 4, 6, 8, 10, 12, and 24 h. The repeatability was determined by preparing six bilberry extracts sample solutions independently and calculating the contents of the Cy-3-glc and the RSD.

## 4. Conclusions

In this study, an easy HPLC fingerprint method, combining reference standard extracts, was set up to authenticate the anthocyanin profile of bilberry extracts. By using this method, one batch of bilberry extract was distinguished from others. In addition, it has been found that bilberry extract is unique for its anthocyanin profile, which is more plentiful than other berry extracts. Besides, the content of anthocyanin Cy-3-glc was determined in each extract. Furthermore, if more standards are used to quantify the reference standard extract of the bilberry in advance, this easy and effective HPLC method could be used for the quality control of bilberry extracts, offering a substitute for the external standard calibration method.

## Figures and Tables

**Figure 1 molecules-25-02514-f001:**
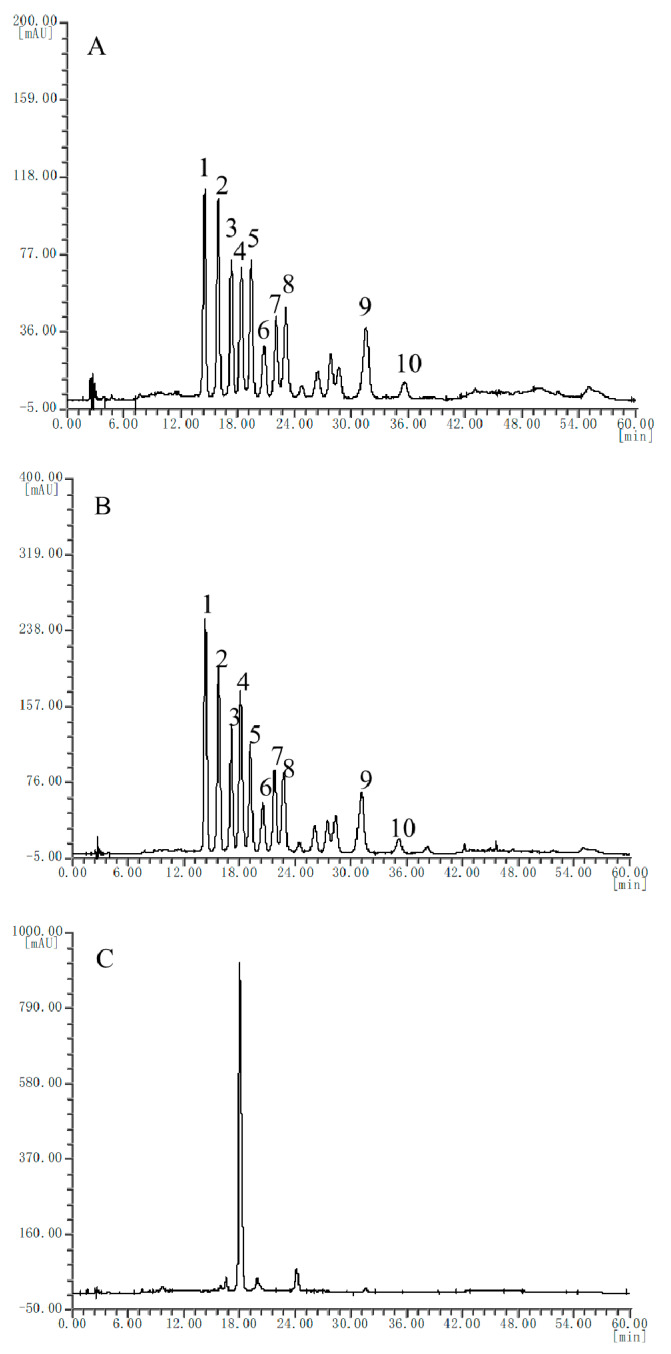
Representative HPLC (high-performance liquid chromatography) chromatograms of the reference extract (**A**), the bilberry extract samples (**B**), and bilberry extract sample A6 (**C**). Peak identification: 1, Delphinidin-3-*O*-galactoside (Del-3-gal); 2, Delphinidin-3-*O*-glucoside (Del-3-glc); 3, Cyanidin-3-*O*-galactoside (Cy-3-gal); 4, Delphinidin-3-*O*-arabinoside (Del-3-ara); 5, Cyanidin-3-*O*-glucoside (Cy-3-glc); 6, Petunidin-3-*O*-galactoside (Pet-3-gal); 7, Cyanidin-3-*O*-arabinoside (Cy-3-ara); 8, Petunidin-3-*O*-glucoside (Pet-3-glc); 9, Malvidin-3-*O*-glucoside (Mal-3-glc); 10, Malvidin-3-*O*-arabinoside (Mal-3-ara).

**Figure 2 molecules-25-02514-f002:**
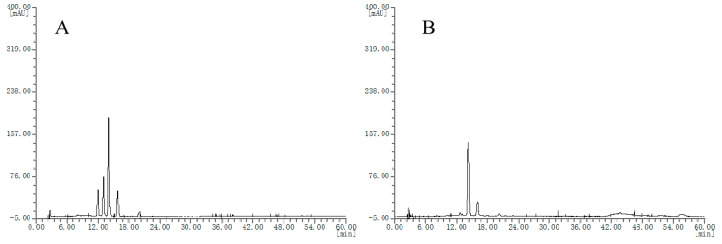
HPLC chromatograms of the blueberry extracts, (**A**), chromatogram type of B2 and B3 samples; (**B**) chromatogram type of B1, B4, B5, B6 and B7 samples.

**Figure 3 molecules-25-02514-f003:**
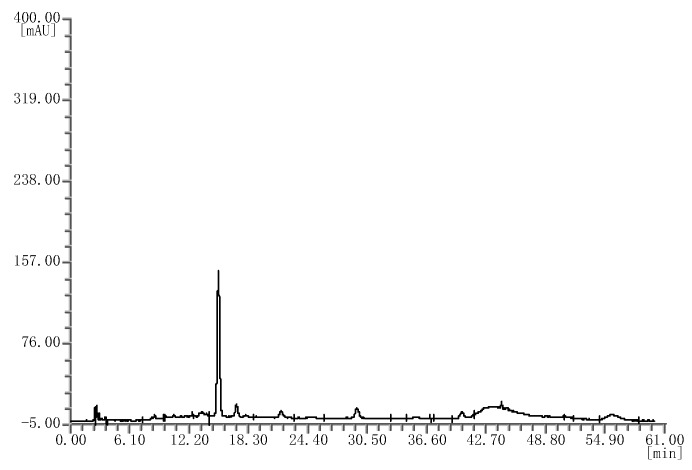
HPLC chromatograms of the mulberry extracts.

**Figure 4 molecules-25-02514-f004:**
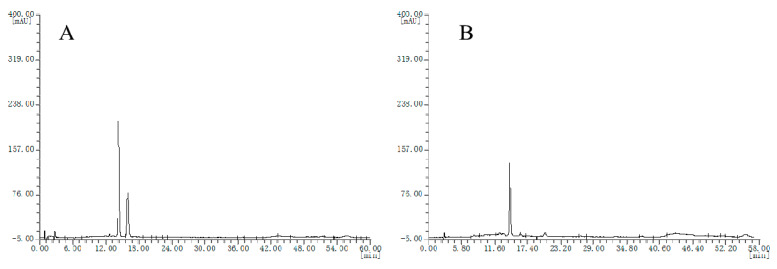
HPLC chromatograms of the cranberry extracts, (**A**), chromatogram type of E1 sample; (**B**) chromatogram type of E2 sample.

**Figure 5 molecules-25-02514-f005:**
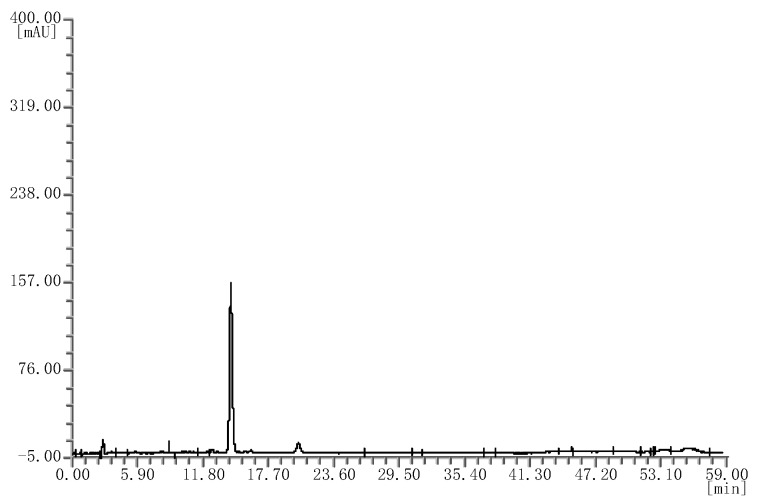
HPLC chromatograms of the black rice extracts.

**Figure 6 molecules-25-02514-f006:**
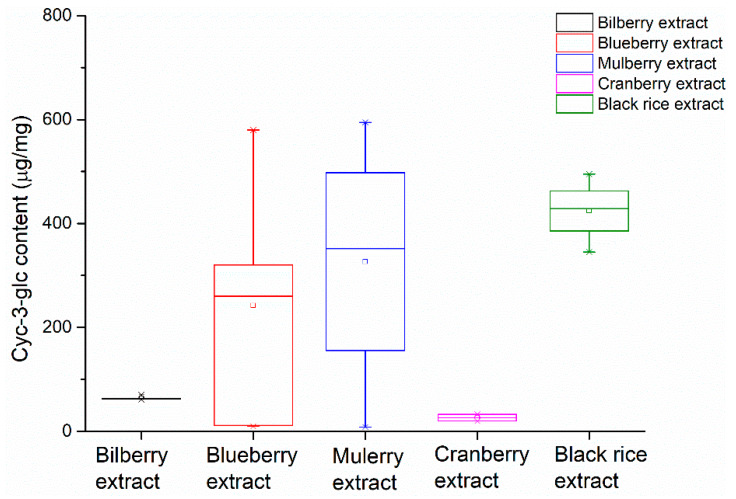
Comparison of the contents of cyanidin-3-*O*-glucoside in different plant extracts.

**Table 1 molecules-25-02514-t001:** Separation resolutions of the compounds under different elution programs.

Compound	P1	P2	P3	P4	P5	P6
Delphinidin-3-*O*-galactoside (Del-3-gal)	6.44	17.93	6.94	6.84	7.53	8.35
Delphinidin-3-*O*-glucoside (Del-3-glc)	4.45	3.11	1.55	2.85	2.84	2.93
Cyanidin-3-*O*-galactoside (Cy-3-gal)	2.11	2.38	2.66	2.64	2.65	2.63
Delphinidin-3-*O*-arabinoside (Del-3-ara)	2.72	1.63	1.61	1.72	1.76	1.84
Cyanidin-3-*O*-glucoside (Cy-3-glc)	2.43	1.57	1.59	1.74	1.77	1.83
Petunidin-3-*O*-galactoside (Pet-3-gal)	2.55	1.61	1.79	1.97	2.11	2.20
Cyanidin-3-*O*-arabinoside (Cy-3-ara)	1.11	-	2.20	1.98	1.94	1.88
Petunidin-3-*O*-glucoside (Pet-3-glc)	2.03	-	0.68	1.07	1.33	1.54
Malvidin-3-*O*-glucoside (Mal-3-glc)	2.13	1.78	1.40	2.55	2.75	2.86
Malvidin-3-*O*-arabinoside (Mal-3-ara)	1.41	2.94	1.58	2.34	4.33	7.23

**Table 2 molecules-25-02514-t002:** The retention time (t_R_), peak area (PA), relative peak relative retention time (RRT), and relative peak area (RPA) of 10 characteristic peaks in the bilberry reference extract and bilberry extract samples (*n* = 3).

Component ^a^	t_R_ (min) ^b^	RRT		Peak area (mAU) ^b^	RPA	
		Average	RSD (%)		Average	RSD (%)
1	14.68	0.75	0.58	2065.633	1.15	0.53
2	15.94	0.82	0.17	2514.939	1.34	5.69
3	17.14	0.89	0.30	1686.999	0.89	6.40
4	17.95	0.94	0.28	1616.961	1.03	4.66
5 (S)	18.80	1.00	0	1864.444	1.00	4.36
6	19.80	1.07	0.10	813.383	0.43	7.25
7	20.84	1.13	0.22	1224.583	0.71	6.03
8	21.43	1.19	0.09	1528.597	0.87	3.96
9	27.09	1.63	0.93	1599.29	0.88	0.11
10	29.88	1.84	1.19	261.977	0.20	40.03

^a^ Component: 1, Del-3-gal; 2, Del-3-glc; 3, Cy-3-gal; 4, Del-3-ara; 5, Cy-3-glc; 6, Pet-3-gal; 7, Cy-3-ara; 8, Pet-3-glc; 9, Mal-3-glc; 10, Mal-3-ara. ^b^ Retention times and peak areas of the ten peaks in bilberry reference extract.

**Table 3 molecules-25-02514-t003:** Stability, precision, and repeatability of the Cy-3-glc quantification method.

Item	Stability (*n* = 1)	Inter-Day Precision (*n* = 1)	Intra-Day Precision (*n* = 1)	Repeatability (*n* = 6)
RSD (%)	0.34	0.30	0.90	0.98

**Table 4 molecules-25-02514-t004:** The recovery rate of the Cy-3-glc in bilberry extract.

No.	Original (μg)	Added (μg)	Found (μg)	Recovery yield (%)	RSD (%)
1	505.13	479.04	988.04	100.81	1.29
2	505.86	479.04	998.14	102.76	
3	506.42	479.04	981.56	99.19	
4	505.94	479.04	983.62	99.72	
5	505.41	479.04	985.15	100.15	
6	505.78	479.04	982.97	99.61	

**Table 5 molecules-25-02514-t005:** Contents of Cy-3-glc in bilberry extract samples (*n* = 3).

Sample No.	*C*_m_ (μg/mL)	*C*_x_ (μg/mL)	*η* (μg/mg)
A1	1504	97.792	65.021
A2	1513	115.54	76.365
A3	1512	98.380	65.066
A4	1509	93.638	62.053
A5	1504	98.175	65.276

**Table 6 molecules-25-02514-t006:** Information about the extract samples.

No.	Extract kind	Batch No.	Manufacture
Bilberry dry extract CRS	Bilberry reference standard extract	−	J&K Scientific Ltd.
A1	Bilberry extract	161004	Longze Biotech. Co., Ltd.
A2	Bilberry extract	20160709	Luzhijian Biotech. Co., Ltd.
A3	Bilberry extract	−	Anhui, Shangshan Biotech. Co., Ltd.
A4	Bilberry extract	B1161101	Xian, Haotian Biotech. Co., Ltd.
A5	Bilberry extract	161222	Zhejiang, Huisong Biotech. Co., Ltd.
A6	Bilberry extract	160725	Shanxi, Baicaocui Biotech. Co., Ltd.
B1	Blueberry extract	161208	Shanxi, Haolin Biotech. Co., Ltd.
B2	Blueberry extract	20161126	Longze Biotech. Co., Ltd.
B3	Blueberry extract	Bl161202	Xian, Haotian Biotech. Co., Ltd.
B4	Blueberry extract	Rm161025	Baoji, Runmu Biotech. Co., Ltd.
B5	Blueberry extract	Rm161025	Baoji, Runmu Biotech. Co., Ltd.
B6	Blueberry extract	−	Shanxi, Baicaocui Biotech. Co., Ltd.
B7	Blueberry extract	−	Shanxi, Longfu Biotech. Co., Ltd.
C1	Mulberry extract	Rm161007	Baoji, Runmu Biotech. Co., Ltd.
C2	Mulberry extract	Rm161007	Baoji, Runmu Biotech. Co., Ltd.
C3	Mulberry extract	150523	Huzhou, Liuyin Biotech. Co., Ltd.
C4	Mulberry extract	160819	Shanxi, Baicaocui Biotech. Co., Ltd.
D1	Cranberry extract	Rm161025	Baoji, Runmu Biotech. Co., Ltd.
D2	Cranberry extract	−	Shanxi, Longfu Biotech. Co., Ltd.
E1	Black rice extract	160623	Shanxi, Haolin Biotech. Co., Ltd.
E2	Black rice extract	HM20160407/s	Hubei, Zixin Biotech. Co., Ltd.
E3	Black rice extract	−	Anhui, Shangshan Biotech. Co., Ltd.
E4	Black rice extract	BR161001	Xian, Haotian Biotech. Co., Ltd.

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
