# Peer review of "Authentication of the Bilberry Extracts by an HPLC Fingerprint Method Combining Reference Standard Extracts"

_molecules, 2020, doi:10.3390/molecules25112514_

Round 1
Reviewer 1 Report
Authors have developed a chromatographic method for the authentication of bilberry extracts. The study is clear, direct and focused. Manuscript is well written and presented.
I have only minor points and suggestions.
- Please presented a superimposed chromatogram of all reference extracts, i.e., bilberry, blueberry, mulberry and black rice.
- Please mention in the conclusion that, by using the currently proposed method, the comercial bilberry tested extracts were all different from the reference extract. This must call the producers attention to a better fruit/plant processing before marketing them.
Minor:
Authors must check the manuscript for uniformity... Sometimes blueberry is being used instead of bilberry. This is sometimes confusing for the reader.
Item 3.2.1. Standards preparation... I don't think authors prepared a standard from 'approximately' 10 mg... Please correct. The same applies for the next topic.
Reviewer 2 Report
The manuscript describes the setup of a HPLC method for bilberry extract fingerprinting intended for authentication and quality control procedures.
The manuscript is fairly written and the procedure sufficiently detailed and sound. However, a revision is needed since there is a major issue concerning the absence of any statistical analysis of data both for qualitative (i.e. fingerprinting) or quantitative results. the manuscript has also some minor issues concerning the missing of some important information which can greatly improve the manuscript quality. In detail:
Major remarks
- In section 2.3 the authors wrote “By comparing the HPLC chromatogram of blueberry extract with those of blueberry, mulberry, cranberry and black rice extracts, it can be concluded that the anthocyanin profile of bilberry extract is quite different from that of the other berry extract”. However, there is no information about the comparison criteria or protocols. Even if the chromatograms of extracts from different origin look different, classification and comparison cannot rely on subjective evaluations. Nowadays there are many statistic classification methods that can give an objective and impacting result visualization. I suggest the author to use a unsupervised multivariate statistic approach such as PCA, which should give a nice data interpretation showing similarities among samples of the same type/origin and what are the main features allowing to distinguish samples from different origin. At a second stage a supervised method such as PLS-DA can also be applied for building a predictive model able to assign any unknown sample to a category.
- Whiskers plot would be much clear for figure 6 to show the quantitative differences between the extracts. As from qualitative data (fingerprinting) there is no statistic data analysis to validate whether extracts from different origins have different contents or content variabilities for Cy-3-glc.
Minor remarks
- Concerning the validation of the method for quantitative measurement of Cy-3-glc, the authors did not indicate which guideline was considered for validation protocol. I suggest the authors to follow and cite the official guideline of the regulatory institution which best pertain the intended use of the method (e.g. FDA guidelines are a reasonable choice for method intended for monitoring food or food extracts)
- Concerning validation, LOD vas reported but not the lowest limit of quantitation (LLOQ) which is similarly important to allow interpretation of the data
- the matrix used for building the calibration curve for Cy-3-glc quantitation was quite different compared to any extract. This can trigger matrix effect issues impacting on method accuracy. Did the author consider the hypothesis to use the standard addition method to generate a calibration curve directly into an extract allowing the quantification of the intrinsic amount of Cy-3-glc?
- In section 2.1 the authors wrote “the separation resolution of all the anthocyanins was the best under elution program 6” but no data about peak resolution as calculated from the chromatograms was given. Was peak resolution the only criterion for the choice of the method? A reasonable choice would be to use the method allowing the shortest analysis time but still maintaining an acceptable peak resolution (Rs>1) for any adjacent peak pair selected from the chromatogram
- HPLC fingerprinting criteria are not reported. There is no information whether compound identification relied on retention time/elution order only or other control levels were included in the fingerprinting protocol (e.g. determination of UV peak purity). In section 2.1 the authors wrote “10 characteristic peaks were identified according to the instruction book of the reference extract” but the reference to the instruction is missing. If it is not possible to cite or refer to a specific document, the main instruction given for peak identification must be reported in the manuscript
- Table 4 should report the replicate number (N) originating the reported RSDs
- Chromatograms are blurry and especially figure 3 is not readable. I suggest the authors to improve the quality of the chromatograms
Reviewer 3 Report
In this manuscript, the authors have developed simple HPLC fingerprint method for identification of the bilberry extract. Other berry extracts: blueberry extracts, mulberry extracts, cranberry extracts and black rice extracts were also analyzed by HPLC. The structure of the article is rather well organized and well balanced. The drawings are legible and well described.
However, I do not recommend to publish the manuscript in Molecules because of the major concerns that have arisen during its reading:
1. The lack of novelty. There is a number of published HPLC methods to authenticate bilberry extracts and detect adulteration. The authors cite only some of them in the introduction. Current pharmacopeias and some another papers provide HPLC anthocyanin profiles to identify commercial bilberry extracts. Nevertheless, the authors did not provide the comparison of the proposed method with already known ones. What is the novelty and advantages of this work compared to papers published earlier?
2. The identification procedure may not be sufficient enough to distinguish genuine bilberry extracts from adulterated material. This is primarily due to the noninclusion of compositional variations in bilberry when sourced from different geographical regions.
3. Validation of the HPLC quantification method: the lack of the limit of quantification (LOQ)
4. Table 3: The RPA data were not presented correctly
Additionally:
1. Subsection 2.2- first sentence- Shouldn't there be figure 1B and figures S7-S12?
2. Subsection 2.5 – concentration of Cy-3-glc in the bilberry extract is given in units mg/mL (in text), should be in μg/mL, in addition, in the text it is marked with the symbol Cs but in the table Cx
3. Preparation of sample solution is described for bilberry extract. What about other extracts?
4. Page 9 – there is probably mistake in the sentence: ,,Then four batches of blueberry extracts, batches of bilberry extracts, and four batches of blueberry extracts were analyzed, and their HPLC chromatograms were compared’’ (probably ,,four batches of blueberry’’ was repeated, maybe another extracts?)
5. There are some typos in the text. The text of manuscript should be carefully checked.
Round 2
Reviewer 2 Report
The authors addressed all the issues found in the first draft. The manuscript can now be published as is.
Reviewer 3 Report
Authors have changed the manuscript accordingly and now it looks better. The novelty of work was explained and highlighted as well as proposed method was compared with already known ones. So I decided to recommend the manuscript for publication in present form.